# Herbicide risk assessments of non-target terrestrial plant communities: A graphical user interface for the plant community model IBC-grass

Jette Reeg[1]*, Simon Heine[2], Christine Mihan[2], Sean McGee[3], Thomas G. Preuss[2], Florian Jeltsch[1,4]

1 Plant Ecology and Nature Conservation, University of Potsdam, Potsdam, Germany, 2 Bayer AG, Monheim am Rhein, Germany, 3 Bayer CropScience, Research Triangle Park, North Carolina, United States of America, 4 Berlin-Brandenburg Institute of Advances Biodiversity Research, Berlin, Germany

* jreeg@uni-potsdam.de

**Data Availability Statement:** All software files and additional information are available at https:// github.com/JetteReeg/IBCgrassGUI. Additional

## Abstract

Plants located adjacent to agricultural fields are important for maintaining biodiversity in semi-natural landscapes. To avoid undesired impacts on these plants due to herbicide application on the arable fields, regulatory risk assessments are conducted prior to registration to ensure proposed uses of plant protection products do not present an unacceptable risk. The current risk assessment approach for these non-target terrestrial plants (NTTPs) examines impacts at the individual-level as a surrogate approach for protecting the plant community due to the inherent difficulties of directly assessing population or community level impacts. However, modelling approaches are suitable higher tier tools to upscale individual-level effects to community level. IBC-grass is a sophisticated plant community model, which has already been applied in several studies. However, as it is a console application software, it was not deemed sufficiently user-friendly for risk managers and assessors to be conveniently operated without prior expertise in ecological models. Here, we present a user-friendly and open source graphical user interface (GUI) for the application of IBC-grass in regulatory herbicide risk assessment. It facilitates the use of the plant community model for predicting long-term impacts of herbicide applications on NTTP communities. The GUI offers two options to integrate herbicide impacts: (1) dose responses based on current standard experiments (acc. to testing guidelines) and (2) based on specific effect intensities. Both options represent suitable higher tier options for future risk assessments of NTTPs as well as for research on the ecological relevance of effects.

## Introduction

Agricultural land covers more than half of the terrestrial landscape in Europe. Especially in intensively managed croplands, agricultural practices may lead to undesired impacts on semi-natural landscape structures such as field boundaries or hedgerows. For example depending

relevant data are within the paper and the
Supporting Information file, except for the
confidential dose-response data used in the
supporting information file.

**Funding:** The project was funded by Bayer AG.
Authors employed by Bayer AG worked on
preparing this manuscript (see Author
contributions). We acknowledge the support of the
Deutsche Forschungsgemeinschaft and Open
Access Publishing Fund of University of Potsdam.

**Competing interests:** The project was funded by
Bayer AG. Authors employed by Bayer AG, namely
SH, CM, TP, SM, worked on preparing this
manuscript (see Author contributions). This does
not alter our adherence to PLOS ONE policies on
sharing data and materials.

on wind conditions, the application of herbicides on crop fields may reach non-target areas such as field boundaries through drift. This can lead to unintended effects on non-target plant communities. As semi-natural landscapes play an important role in maintaining biodiversity in agricultural landscapes, for instance by providing food or serving as shelter habitat [1,2], there is a need for protecting these field margins from unacceptable adverse impacts caused by agricultural practices. Therefore, before a new product is placed on the market, a risk assessment is conducted to evaluate the potential risk of applying the product according to the proposed label specifications [3].

The current risk assessment scheme for non-target terrestrial plants (NTTPs), i.e. plants not intended to be affected by the plant protection product, follows a tiered approach. The baseline risk assessment is based on standardized greenhouse experiments to test for impacts on vegetative vigour and seedling emergence at the individual plant level under different application rates (OECD testing guidelines 208 and 227 [4,5]). These studies are not designed to cover inter- and intraspecific competition between plant individuals in a plant community. In general, assessment factors are applied to account for uncertainties such as the extrapolation from greenhouse to the field or the existence of even more sensitive species. Guidance on harmonized and fully accepted higher tier studies, which could potentially overcome some of that uncertainty, is not available.

Modelling approaches are often mentioned as suitable tools for higher tier evaluations [6]. Ecological models overcome the spatial and temporal limitations as well as high resource requirements of empirical field studies. Thus, a range of different environmental conditions can be tested as a full factorial design. While relevant stakeholders (e.g., risk assessors and risk managers) have expert knowledge with the ecological aspects and empirical data used in ecological modelling, they may lack experience with computational aspects of models. Therefore ecological models need to fulfil certain requirements to be considered as suitable higher tier approaches: (1) comparison of model predicted effects against empirically measured data to increase the credibility of the simplified model to realistically reflect herbicide impacts [7, 8]; (2) a sensitivity analyses of model parameters that are not based on empirical data need to show the robustness of the model; (3) a comprehensive model documentation should facilitate the communication between model developers and users, e.g. regulators, by presenting the applicability and capabilities of the model [9,10]. These requirements are essential to establish trust in the models' capabilities but also to reveal possible limitations that come along with the simplification of the real ecological system.

The plant community model IBC-grass (Individual-Based plant Community model for GRASSlands) represents such a suitable approach to extrapolate individual-level effects measured in standard guideline studies [5,11] to plant populations in community context. Recent studies highlighted the capability of the model to detect herbicide induced impacts on plant communities [12], showed different sensitivities of important plant attributes [13] and validated IBC-grass against short-term [14] and long-term empirical data (see supporting information file S1 File). Model development is documented using the ODD protocol (Overview, Design concept and Details [15]).

Although all requirements mentioned earlier are fulfilled, the model was, up to now, not convenient to use as it was developed as a console application, which would likely lead to hesitation to apply the model, especially for researchers not trained in modelling. Graphical user interfaces (GUI) are suitable tools to facilitate the application by guiding the user through settings and analyses of a simulation model. The objective of this paper is to present a graphical user interface (GUI) for IBC-grass that facilitates the use of the model for risk assessment purposes without requiring any programming skills.

## Methods

### The model IBC-grass

Fig 1 presents the flowchart of the processes integrated in IBC-grass. In the following we will only give a broad overview of the main principles of IBC-grass. A detailed model documentation including the description of the underlying functions and processes can be found in the software package (ODD and GMP documents in [16]).

**Trait-based approach.**   Plant trait characteristics are known to influence plant community dynamics [17]. Species with similar set of trait characteristics, a plant functional type (PFT), are known to respond similar to environmental conditions. Thus, the PFT approach can be used to make conclusions for several plant communities consisting of different plant species but similar PFTs. In IBC-grass plant species are classified into plant functional types (PFTs) according to selected trait characteristics (Table 1) known to be important for population dynamics. They include several trade-off or correlations, e.g. seed mass and plant mass. Three different trait data bases are used to collect the corresponding trait values and to classify plant species into the different PFTs [18–20]. Each PFT is assigned a specific identification acronym, which consists of the major trait characteristics (Table 2). The GUI gives examples for each PFT ID.

**2-layer zone of influence approach.**   To account for competition between plant individuals, a 2-layer zone of influence approach is implemented for the above- and belowground compartments: Depending on the growth form, plant size and specific leaf area/root area, each plant individual has a specific zone of influence, a circular area in which it takes up resources (Fig 2). In overlapping zones of influences, plant individuals compete for resources; intraspecific competition being stronger than interspecific competition. Aboveground, only the size asymmetric competition for light is considered: taller plants with an erect growth form shade smaller plants growing as a rosette and thus acquire more resources [21]. For a model being a simplified version of the real world, belowground competition, on the other hand, is assumed to be size symmetric: resource distribution between plant individuals with overlapping zones of influences is independent of the root growth form and only depend on the root mass.

**Spatial and temporal dimensions.**   IBC-grass is a spatially explicit model: it simulates plant community dynamics on a grid 1x1 $cm^2$ cells, representing a patch in a landscape. The size of the grid can vary between 100x100 $cm^2$ (= 1 $m^2$) and 173x173 $cm^2$ (3 $m^2$). The local patch is simulated as a torus, i.e. the edges are connected to each other. Temporal dynamics are simulated in weekly time steps with only the growing period of spring to autumn is considered. During the winter period, a winter dieback of shoot mass and winter mortality is simulated.

## Results

### The graphical user interface

The graphical user interface (GUI) facilitates the application of IBC-grass in herbicide risk assessments. The GUI guides the user through the environmental settings and herbicide settings of IBC-grass, and analyses the results of a set of simulations (see Table 3 for an overview of the model parameters addressed in the GUI). It is an open access software hosted on GitHub [23]. The GUI itself is written in R using the R package RGtk2 [24]. The package includes a folder with the C++ source code files of the plant community model IBC-grass and a folder including the model documentation (ODD protocol [14] and GMP document [8]) and the detailed manual for the GUI. Here, we will only give a summary of the GUI. For detailed information, please have a look at the manual [16].

**Requirements.**   To run the GUI the following software needs to be installed on the local machine:

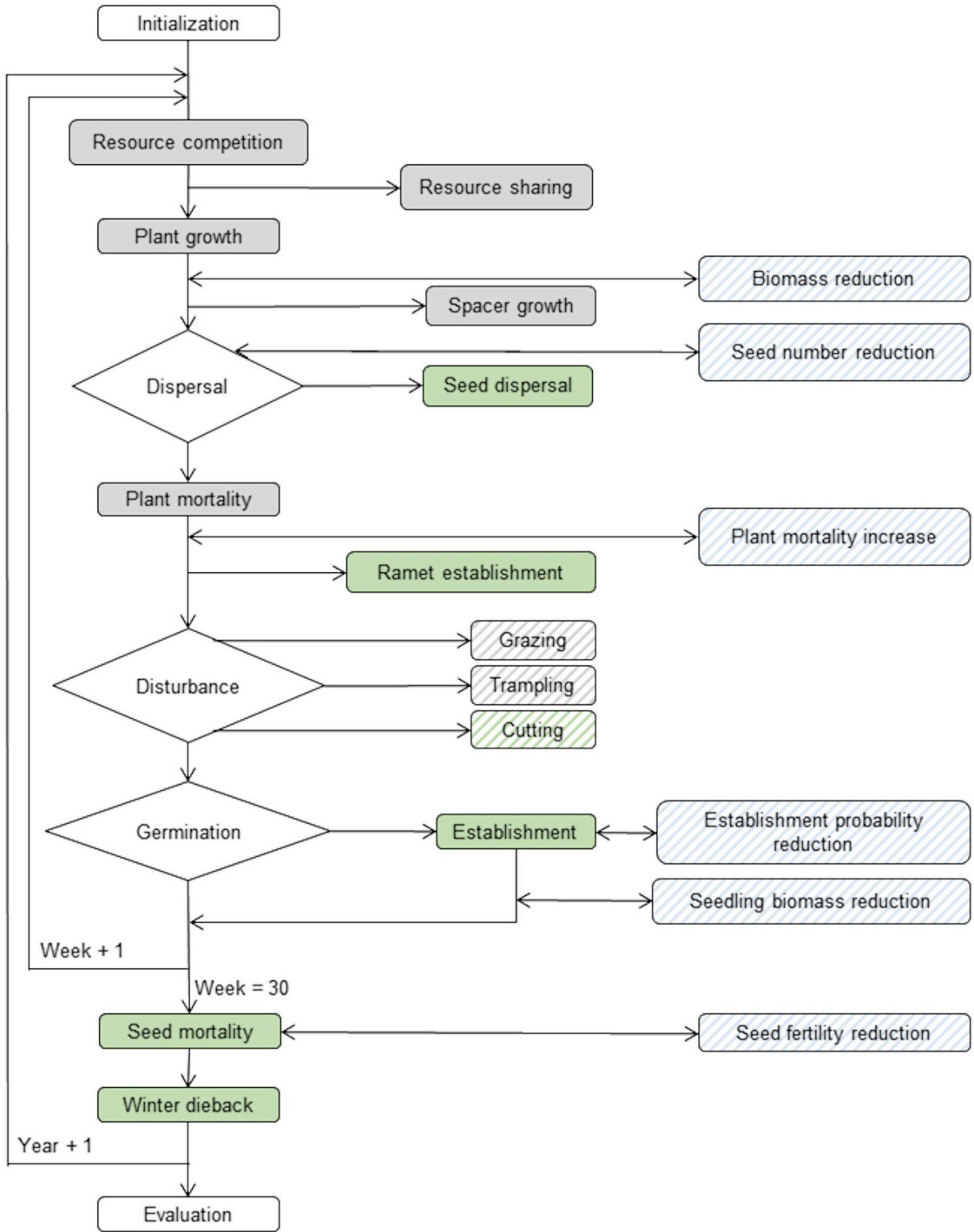

**Fig 1. Flow chart of the processes simulated in IBC-grass.** Grey boxes indicate processes occurring in each simulated week, green boxes indicate processes occurring only in specific weeks and blue boxes indicate potential herbicide-induced effects that the user can turn on or off. Striped boxes indicate processes the user can adjust and change.

**Table 1. Classification of plant species into plant functional types (PFTs).**

| TRAIT | VALUES | BASED ON DATABASE TRAIT | CORRESPONDING MODEL PARAMETERS | | |
|---|---|---|---|---|---|
| **Plant size** | | *seed releasing height*[1] | *maximal plant mass* | *seed mass* | *seed dispersal* |
| | small | < = 0.42 | 1000 mg | 0.1 mg | 0.6 m |
| | medium | 0.42–0.87 | 2000 mg | 0.3 mg | 0.3 m |
| | tall | >0.87 | 5000 mg | 1 mg | 0.1 m |
| **Growth form** | | *rosette attribute*[2] | *leaf mass ratio* | | |
| | erect | erect | 0.5 | | |
| | semi-rosette | semi-rosette | 0.75 | | |
| | rosette | rosette | 1.0 | | |
| **Resource response** | | *ecological strategy after Grime*[2, 3] | *maximal resource units* | *maximal survival under resource stress* | |
| | stress-tolerator | sr, cs, s | 20 | 6 | |
| | intermediate | csr, r | 40 | 4 | |
| | competitor | c, cr | 60 | 2 | |
| **Grazing response** | | *grazing tolerance*[2,4] | *palatability* | *specific leaf area* | |
| | tolerator | 4–6 (resprouter) | 1.0 | 1.0 | |
| | intermediate | 1–3 (no adaption) | 0.5 | 0.75 | |
| | avoider | 7–9 (defence strategies) | 0.25 | 0.5 | |
| **Clonal type** | | *clonality*[5] | *spacer length* | *resource sharing* | |
| | long internodes with resource sharing | lateral spread 0.01–0.25 m/y with persistence of connection | 17.5 cm | 1 | |
| | long internodes without resource sharing | lateral spread 0.01–0.25 m/y with persistence of connection | 17.5 cm | 0 | |
| | short internodes with resource sharing | lateral spread < 0.01m/y with persistence of connection | 2.5 cm | 1 | |
| | short internodes without resource sharing | lateral spread < 0.01m/y with persistence of connection | 2.5 cm | 0 | |
| **Flowering type** | | *symphenological groups*[2, 6] | *start of seed pro-duction* | *end of seed production* | |
| | early | 1–6 | week 1 | week 5 | |
| | late | 7–10 | week 16 | week 20 | |
| **Germination periods** | | | establishment period | | |
| | spring | | weeks 1–4 | | |
| | summer | | weeks 21–25 | | |
| | spring and summer | | week 1–4 and 21–25 | | |
| **Life span** | | *life span*[2] | Maximal plant age | | |
| | annual | a | 1 year | | |
| | perennial | p | 100 years | | |

[1][18]

[2][19]

[3]s: stress tolerator, r: ruderal, c: competitor and combinations thereof

[4]Ordinal scale (9 levels) ranging from 1 (intolerant to grazing) to 9 (very tolerant to grazing)

[5][20]

[6]Ordinal scale (11 levels) giving the time of flowering: 1 (pre spring) to 10 (autumn).

0 –not available

**Table 2. Compilation of the specific PFT ID according to the major traits.**

| PLANT SIZE | GROWTH FORM | RESOURCE RESPONSE TYPE | GRAZING RESPONSE TYPE | CLONAL TYPE | LIFE SPAN | FLOWERING PERIOD | GERMINATION PERIOD |
|---|---|---|---|---|---|---|---|
| Small | Erect | Competitor | Avoider | cl1 short internodes, resource sharing | perennial | early | early |
| Medium | Semi-rosette | Stress-tolerator | Intermediate | cl2 short internodes, no resource sharing | annual | late | late |
| Large | Rosette | Intermediate | Tolerator | cl3 long internodes, resource sharing | | | both |
| | | | | cl4 long internodes, no resource sharing | | | |

- G++ compiler (e.g., MingGW compiler [25]), set as environmental variable.

- In some cases you might need to install GTK+ 3 [26] on your own. However, the GUI will at least try to install it on windows systems.

The GUI was tested under Windows (7 and 10).

**Regional PFT pool.** The GUI includes three regional PFT communities: a common field edge community with high resource input, medium trampling events and one mowing per year; *Calthion* as a nutrient poor grassland with low disturbances by grazing and trampling and one mowing event per year; and *Arrhenatheretalia* as a nutrient rich grassland with high disturbances by grazing and trampling and three mowing events per year. All three communities were used in an early study by Reeg et al. [12].

In addition to the predefined communities, the user has the possibility to create new plant communities either by selecting plant species from one of the three communities mentioned above, or by classifying new plant species into plant functional types (PFTs, Table 1).

**Environmental settings.** For the predefined plant communities the GUI suggests the environmental settings (abiotic as well as biotic conditions such as resource availability and biotic disturbances) that were applied in Reeg et al. [12]. Nevertheless, the setting can always be adjusted.

*Resource settings.* Resources are given in resource units (ru) per $cm^2$, not defining a specific resource. 40 $ru/cm^2$ represent low resource levels, 100 $ru/cm^2$ represent high resource levels.

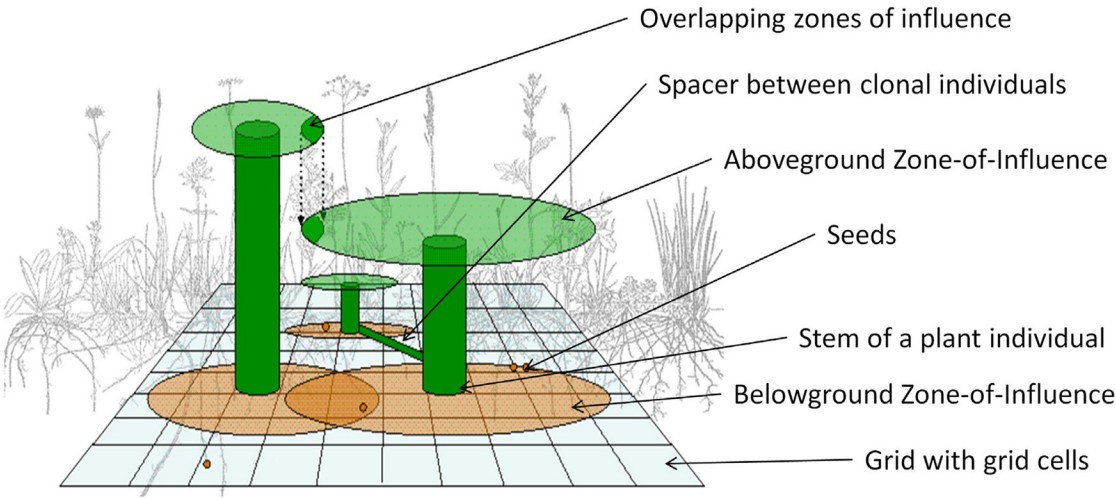

**Fig 2. Graphical scheme of the IBC-grass model.** Graphic is adapted from Weiß [22].

**Table 3. Model parameters addressed in the GUI.**

| IBC-GRASS PARAMETER | EXPLANATION |
|---|---|
| IBCcommunity | PFT community |
| IBCgridsize | Number of grid cells |
| IBCabampl | Amplitude for aboveground resource seasonality |
| IBCabres | Aboveground resource units |
| IBCbelres | Belowground resource units |
| IBCSeedInput | Number of seeds per PFT added at the beginning of each year |
| IBCcut | Number of cutting events per year |
| IBCgraz | Amount of area grazed during one year |
| IBCtramp | Amount of area trampled during one year |
| IBCInit | Number of initial years |
| IBCDuration | Number of year with simulated herbicide application |
| IBCRecovery | Number of years following the herbicide application period |
| IBCweekstart | Calendar week of herbicide application |
| IBCherbeffect | If 'txt-file': herbicide effects are based on a txt-file (predict potential effects) |
|  | If 'dose-response': herbicide effects are based on dose-response data |
| IBCApprateScenarios | Annual application rates for each scenario (if herbicide effects are based on dose-response data) |
| BiomassEff | Is plant biomass affected? |
| EstablishmentEff | Is seed establishment affected? |
| SeedlingBiomassEff | Is seedling biomass affected? |
| SeedNumberEff | Is seed number affected? |
| SeedSterilityEff | Is seed sterility affected? |
| SurvivalEff | Is plant survival affected? |
| IBCrepetition | Number of repetitions |

Resources of the above- and belowground compartment are differentiated. For the aboveground resources a seasonality effect can be included: The aboveground resources can follow a sine curve. The selected amplitude determines the height of the sine curve. This seasonality can be based on light intensity data.

*Disturbance settings.* Three different disturbances are distinguished: grazing, trampling and cutting. Trampling removes the aboveground biomass in x% cells of the grid over the year, with one footprint being reflected as a 10 cm$^2$ patch. Grazing only removes a certain percentage of the aboveground shoot mass of plant individuals depending on the palatability of the PFT. Cutting events can occur 1, 2, 3 times a year or never. In one cutting event, the aboveground biomass in the whole grid is removed to a specific cutting height. The shoot mass left after a cutting event depends on the growth form specific to translate shoot mass removal to a cutting height (see ODD protocol for further details in [16]).

**Herbicide settings.** The user has the option to vary the number of years for the initial phase (without herbicide application to stabilize the plant community), herbicide application phase including the timing of the application and recovery phase (without herbicide application). To distinguish herbicide-induced impacts from ecological impacts induced by the model initialization it is important to start herbicide application after the community dynamics stabilized (~25–50 years). Otherwise herbicide impacts might be shaded by high variation between simulation runs.

IBC-grass does not directly account for different modes of action. However, the user can select six different plant attributes to be affected by the herbicide, namely shoot mass, seedling

shoot mass, survival, establishment, seed sterility and seed number, and vary the sensitivity of the different PFTs to cover for broad spectrum and selective herbicides. With these options, the mode of action can be indirectly addressed by affecting plant attributes.

Ideally, herbicide effects are based on dose responses following the current OECD guidance documents 208 and 227 [5,10]. The user is asked to transfer the results of the experiments (specifically the number of test species and for each selected attribute the test rate and measured data per test species). The GUI will then calculate dose responses by optimizing the parameters *EC50* and slope *b* of the Eq 1 to the empirical data using the Nelder-Mead method [27].

$$Effect\ (Application\ rate) = \frac{Application\ rate^b}{ER50^b + Application\ rate^b} \tag{1}$$

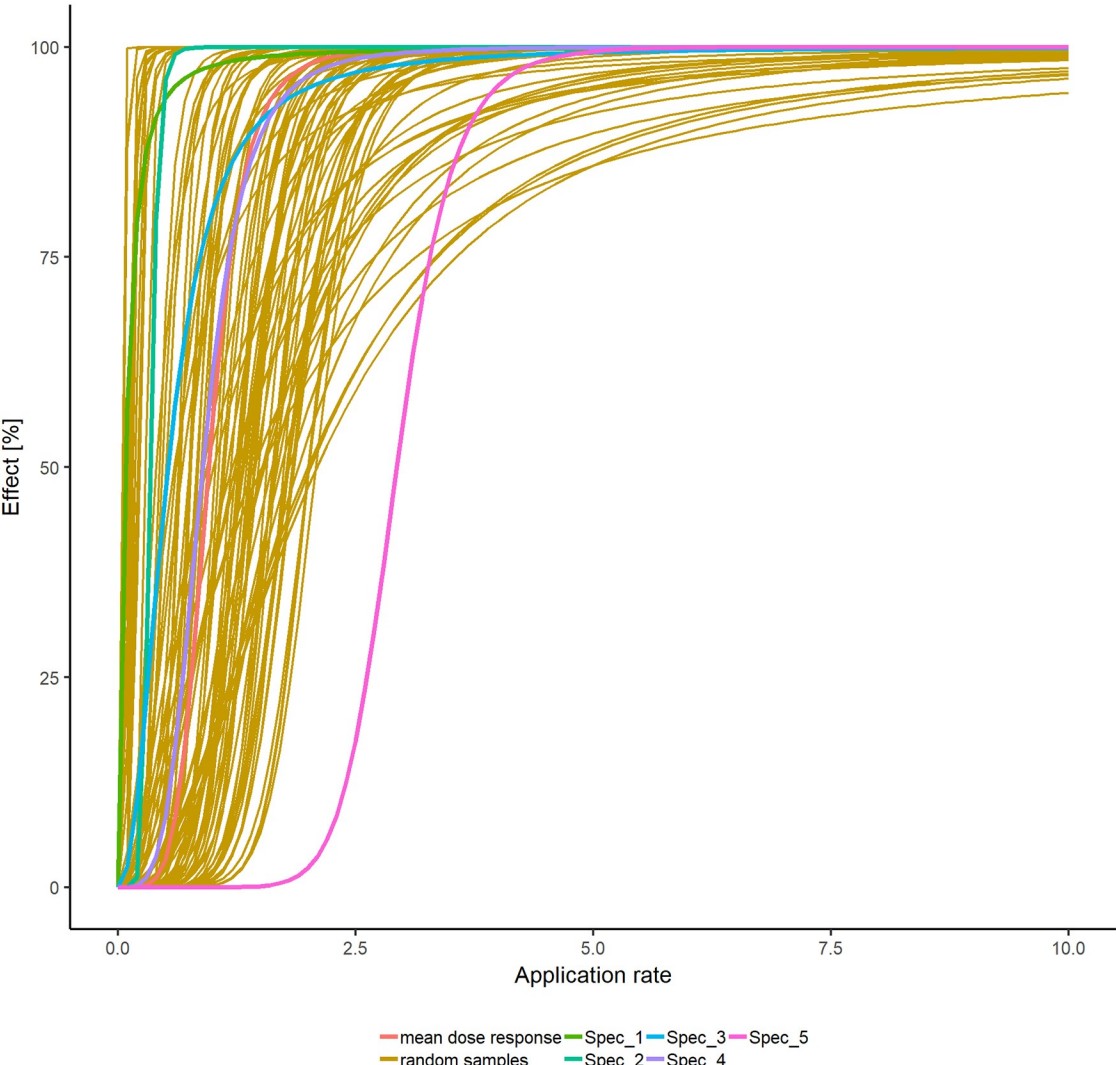

**Fig 3. Example of calculated dose responses based on empirical data of 5 plant species.** Red line represent the mean dose response (with mean of the estimated *EC50* and slope *b*), orange lines represent 100 random dose responses based on the mean and standard deviation of the estimated *EC50* and slope *b*.

However, if the user has no access to dose response data, the GUI offers an alternative approach, which was also used in Reeg et al. [12]: The user can specify effect intensities for each plant attribute in each year of simulated herbicide application. In this way, the user has the possibility to analyse whether a certain individual-level effect intensity has a significant impact on population- and/or community-level.PFT sensitivity.

If the herbicide effects are based on dose-response data, the user needs to assign dose-responses to the PFTs in the plant community model. Either the calculated dose responses can be assigned directly, for instance if a plant species of a certain PFT was tested. However, the sensitivity of many non-crop plant species is still unknown, but the literature review by Christl et al. [28] comparing herbicide sensitivities of crop vs. weed species for various modes of action showed that the range of sensitivities of crop and non-crop species are comparable. Therefore, we suggest assigning random dose responses within the variation of the calculated ones by calculating the mean and standard deviation of both the estimated *EC50* and slope *b* values and then choosing randomly from a uniform distribution within the interval of mean +- standard deviation for each single simulation (Fig 3). In addition to direct and random assignment of dose-responses, a PFT can also be assigned as being not affected at all, for example if a selective herbicide is simulated.

On the other hand, if the user specified only certain effect intensities, the sensitivity of the different PFTs can be assigned as random (0–1), not affected (0), low (0.1–0.35), medium (0.35–0.65), high (0.65–1) or full (1). The effect intensity is multiplied by the random number out of the certain interval. For example, if the effect intensity for plant survival was set to 0.5

**Table 4. Summary of simulation settings for the presented exemplary scenario in which herbicide effects are based on dose response data.** The second exemplary scenario can be found in the software package [16].

| IBC-GRASS PARAMETER | EXEMPLARY SCENARIO |
|---|---|
| **IBCcommunity** | Fieldedge.txt |
| **IBCgridsize** | 173 |
| **IBCabampl** | 0.0 |
| **IBCabres** | 100 |
| **IBCbelres** | 90 |
| **IBCSeedInput** | 10 |
| **IBCcut** | 1 |
| **IBCgraz** | 0.001 |
| **IBCtramp** | 0.1 |
| **IBCInit** | 35 |
| **IBCDuration** | 10 |
| **IBCweekstart** | 11 |
| **IBCRecovery** | 5 |
| **IBCherbeffect** | Dose-response |
| **IBCApprateScenarios** | 1.1 g a.i./ha (herbicide scenario 1), 3.3 g a.i./ha (herbicide scenario 2) [no annual variation] |
| **BiomassEff** | TRUE |
| **EstablishmentEff** | FALSE |
| **SeedlingBiomassEff** | FALSE |
| **SeedNumberEff** | FALSE |
| **SeedSterilityEff** | FALSE |
| **SurvivalEff** | TRUE |
| **IBCrepetition** | 30 |

and the sensitivity of a PFT to low, a random number is drawn between 0.1 and 0.35, e.g. 0.2. The resulting PFT specific effect is 0.1, which means that plant individuals of this PFT have an herbicide-induced mortality probability of 10%.

**Simulation settings.** In a last step, the user specifies the number of repetitions, the simulated plot size, the degree of isolation (as external seed input) and, if herbicide effects are based on dose responses, the number of different herbicide scenarios. The number of repetitions is the number of simulations that have the same model settings in the environmental and herbicide parameters (Monte-Carlo runs).

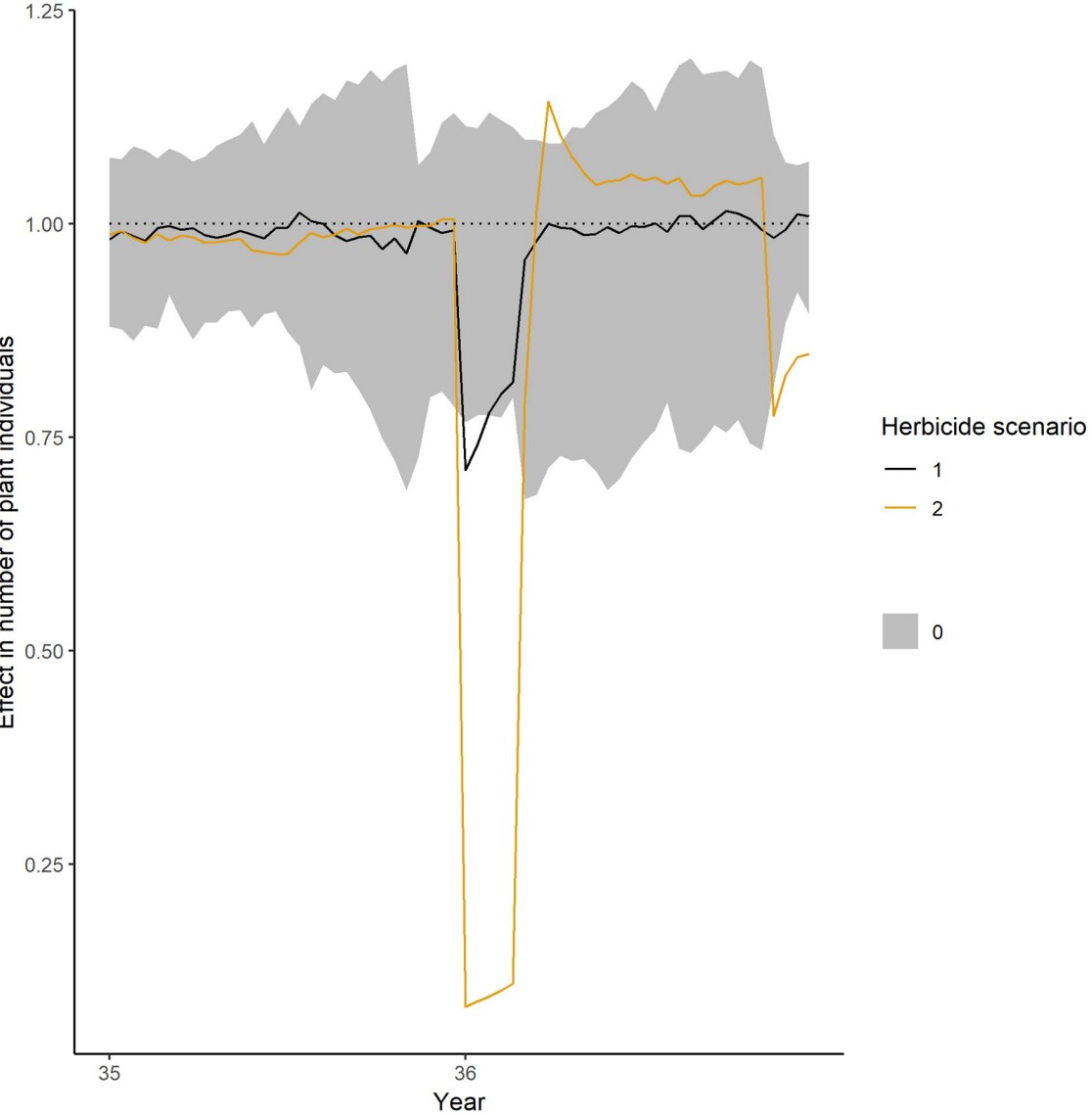

**Fig 4. Short-term impacts on number of plant individuals during the first year of simulated herbicide application.** Values below 1 represent a negative impact, values equal to 1 no impact and values above 1 a positive impact. The theoretical herbicide had an impact on biomass and mortality. PFTs had random dose response-curves. Grey ribbon shows the fluctuation within control simulations, the black line shows the mean for an application rate of 1.1 g a.i./ha (herbicide scenario 1) and the orange line the mean for an application rate of 3.3 g a.i./ha (herbicide scenario 2).

Before the simulations are started, the user needs to specify the annual application rates per scenario. Especially the number of repetitions, the plot size and the number of simulated application rates have a high impact on the running time. To accelerate the running time, the GUI is parallelizing simulations using all cores but two, i.e. if a local machine has four cores, the GUI will use two of them to parallelize the IBC-grass Monte-Carlo simulations.

**Analyses.** Raw output data of one simulation run include responses on population- and community-level in weekly time steps: PFT population size, shoot mass and cover on population-level and number of PFTs, number of individuals, aboveground biomass and four different diversity indices (Evenness, Shannon, Simpson and inverse Simpson) on community-level.

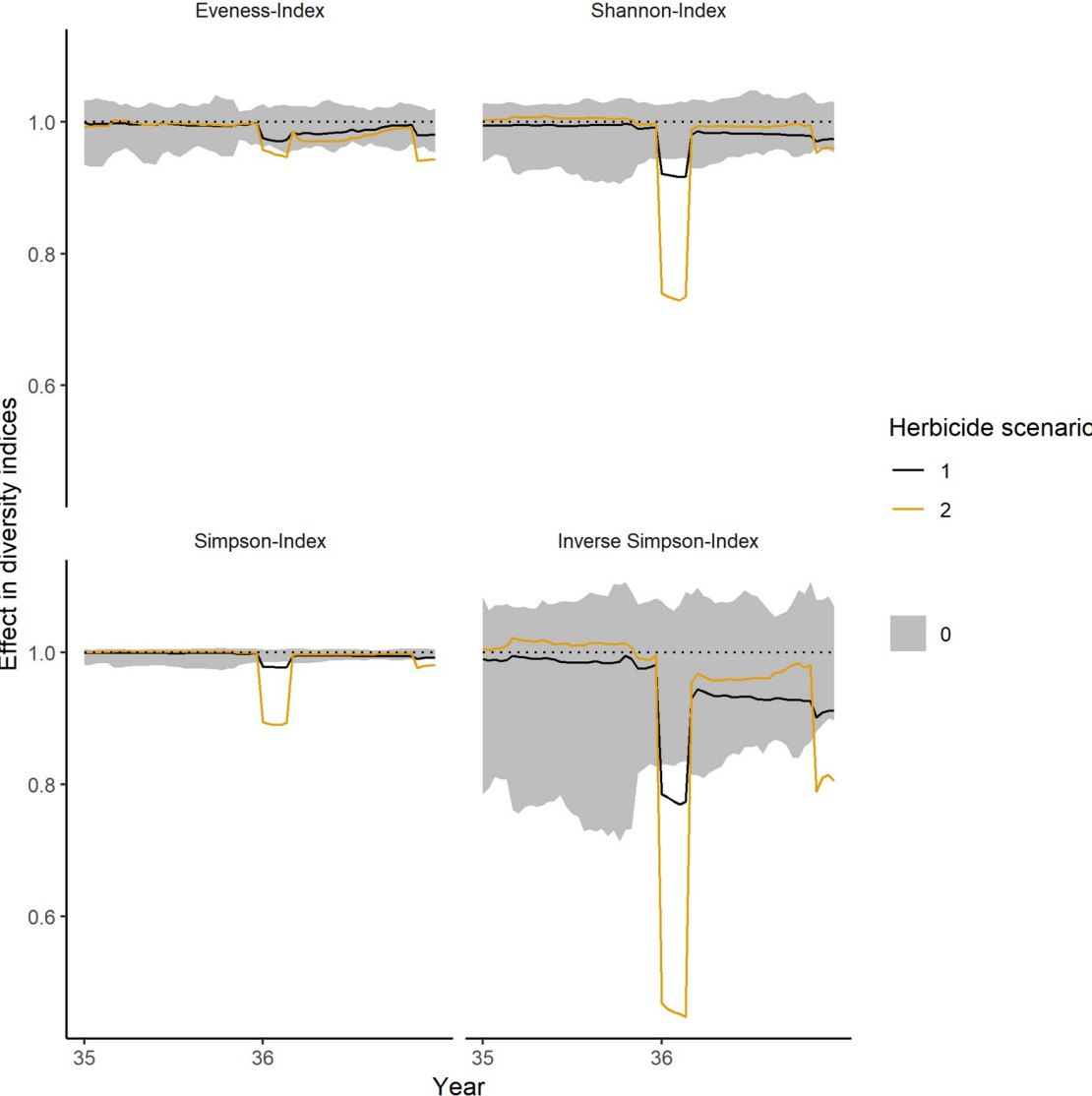

**Fig 5. Short-term impacts on different diversity indices during the first year of simulated herbicide application.** Values below 1 represent a negative impact, values equal to 1 no impact and values above 1 a positive impact. The theoretical herbicide had an impact on biomass and mortality. PFTs had random dose response curves. Grey ribbons show the fluctuations within control simulations, the black lines show the mean for an application rate of 1.1 g a.i./ha (herbicide scenario 1) and the orange lines the mean for an application rate of 3.3 g a.i./ha (herbicide scenario 2).

The GUI will further analyse these data, however, the user can keep the raw data for individual analyses. Note that storage footprint can be very high.

The model output, i.e. the values of the different endpoints for each modelled time step, is standardized by the mean of the corresponding control simulation: For each single Monte Carlo simulation run (control and treatments), the value per time step is divided by the mean of the control of the specific time step to calculate standardized effects relative to the control mean (i.e. a resulting standardized value 0.7 represents a 30% decrease in the specific endpoint compared to the mean of the control–e.g. in biomass). These data are saved in the files '*resultsPFT.txt*' for population-level endpoints and '*resultsGRD.txt*' for community-level

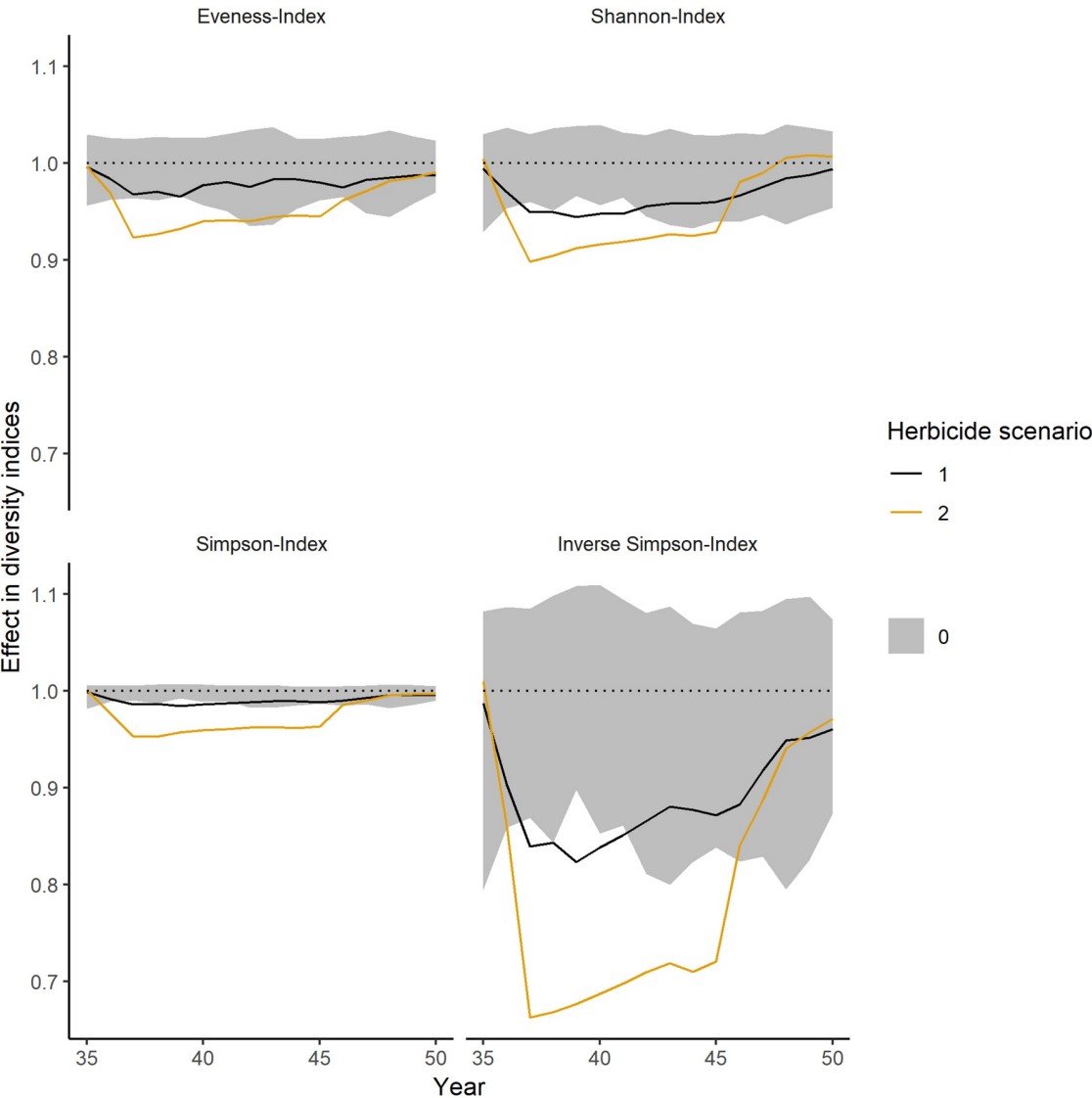

**Fig 6. Long-term impacts on the diversity indices over the simulated herbicide application.** Herbicide application started in year 36 (shown in Fig 5) and ended in year 45. Values below 1 represent a negative impact, values equal to 1 no impact and values above 1 a positive impact. The theoretical herbicide had an impact on biomass and mortality. PFTs had random dose response curves. Grey ribbons show the fluctuations within control simulations (averaged over each year), the black lines show the mean (averaged over each year) of the 1.1 g a.i./ha application rate (herbicide scenario 1), the orange line the mean (averaged over each year) of the 3.3 g a. i./ha application rate (herbicide scenario 2).

endpoints. Further, the effects are averaged (mean (mean effect), 2.5$^{th}$ percentile (maximal effect) and 97.5$^{th}$ percentile (minimal effect)) over all simulations per time step as well as per year. Results are saved as '*effect.timestep.PFT.txt*' and '*effect.year.PFT.txt*' for population level endpoints and '*effect.timestep.GRD.txt*' and '*effect.year.GRD.txt*' for community level endpoints. Based on the weekly analyses ('*effect.timestep.\**'), the number of weeks in which the (mean, minimum, maximum) effect is within a certain interval are summed up. For this, we used effect intervals of <10%, 10–20%, 20–30%, 30–40% and >50%. The results are saved for each endpoint separately as '*\*_PFT.txt*' for population level endpoints and '*\*_GRD.txt*' for community level endpoints. Note that positive effects will be included in the interval <10%.

## Example

The IBC-grass GUI package includes two examples: one for herbicide effects based on dose response data and one for herbicide effects based on specific effect intensities. Here we present selected examples of potential output of the GUI only for the first example. However, as both examples are included in the GUI package, the user can load both projects to look at the full set of results.

**Project settings.**   This example simulated the impact of a potential herbicide for PFT populations of a field edge community. The herbicide effect was simulated for 10 years (1 application/year, in the first week of the growing period) after an initialization phase of 35 years where no application of any herbicide took place. The intention of the initialization phase is to build a stable plant community. The herbicide effects were based on dose-responses for impacts on biomass and mortality of 5 test species. Dose responses were randomly sampled for each PFT in each Monte Carlo simulation. The simulation settings are summarized in Table 4.

**Results.**   *Community-level.* During the first year of herbicide application, the number of individuals is decreasing and exceeding the normal range of fluctuations (Fig 4). The effects are increasing with higher application rates. The pattern is similar for all except of one diversity index: The herbicide scenario with the lower application rate (herbicide scenario 1, 1.1 g a. i./ha) showed no significant impact on the Evenness (i.e. the mean effect is not exceeding the range of the control simulations) (Fig 5).

**Table 5. Number of weeks in which the mean (minimal and maximal) negative effect on the number of plant individuals and the inverse Simpson index is within a certain effect class.** As IBC-grass simulates only 30 weeks of growing period, the maximal number of weeks is 30. The simulated herbicide application started in year 36. Numbers in brackets represent the number of weeks in which the minimal and maximal values are within a certain effect class. Please note, that only negative effects are considered in this table. Positive impacts can be observed in Figs 4–7.

|  | YEAR | APPLICATION RATE | <10% | 10–20% | 20–30% | 30–40% | 40–50% | >50% |
|---|---|---|---|---|---|---|---|---|
| NUMBER OF PLANT INDIVIDUALS | 35 | 0 | 30 (0 1) | 0 (0 22) | 0 (0 6) | 0 (0 1) | 0 (0 0) | 0 (0 0) |
|  |  | 1.1 | 30 (0 4) | 0 (0 19) | 0 (0 6) | 0 (0 1) | 0 (0 0) | 0 (0 0) |
|  |  | 3.3 | 30 (0 0) | 0 (0 22) | 0 (0 5) | 0 (0 3) | 0 (0 0) | 0 (0 0) |
|  | 36 | 0 | 30 (0 1) | 0 (0 3) | 0 (0 23) | 0 (0 3) | 0 (0 0) | 0 (0 0) |
|  |  | 1.1 | 25 (4 0) | 2 (1 14) | 3 (0 10) | 0 (0 2) | 0 (0 4) | 0 (0 0) |
|  |  | 3.3 | 20 (4 9) | 3 (1 11) | 2 (0 2) | 0 (0 3) | 0 (0 0) | 5 (5 5) |
| INVERSE SIMPSON INDEX | 35 | 0 | 30 (0 0) | 0 (0 7) | 0 (0 23) | 0 (0 0) | 0 (0 0) | 0 (0 0) |
|  |  | 1.1 | 30 (0 0) | 0 (0 5) | 0 (0 5) | 0 (0 0) | 0 (0 0) | 0 (0 0) |
|  |  | 3.3 | 30 (0 23) | 0 (0 6) | 0 (0 1) | 0 (0 0) | 0 (0 0) | 0 (0 0) |
|  | 36 | 0 | 30 (0 0) | 0 (0 30) | 0 (0 0) | 0 (0 0) | 0 (0 0) | 0 (0 0) |
|  |  | 1.1 | 25 (5 0) | 0 (3 0) | 5 (0 25) | 0 (0 5) | 0 (0 0) | 0 (0 0) |
|  |  | 3.3 | 21 (4 0) | 3 (0 0) | 1 (0 21) | 0 (3 0) | 0 (2 4) | 5 (0 5) |

Over the long-term, all diversity indices showed significant impacts (i.e. the mean effects that are outside of the control fluctuations) in most years for the second herbicide scenario with an application rate of 3.3 g a.i./ha (Fig 6). However, all indices are able to recover within the 5 years of simulated recovery period. For a lower application rate of 1.1 g a.i./ha (herbicide scenario 1), the impact is considerably lower with only a few years of significant effects.

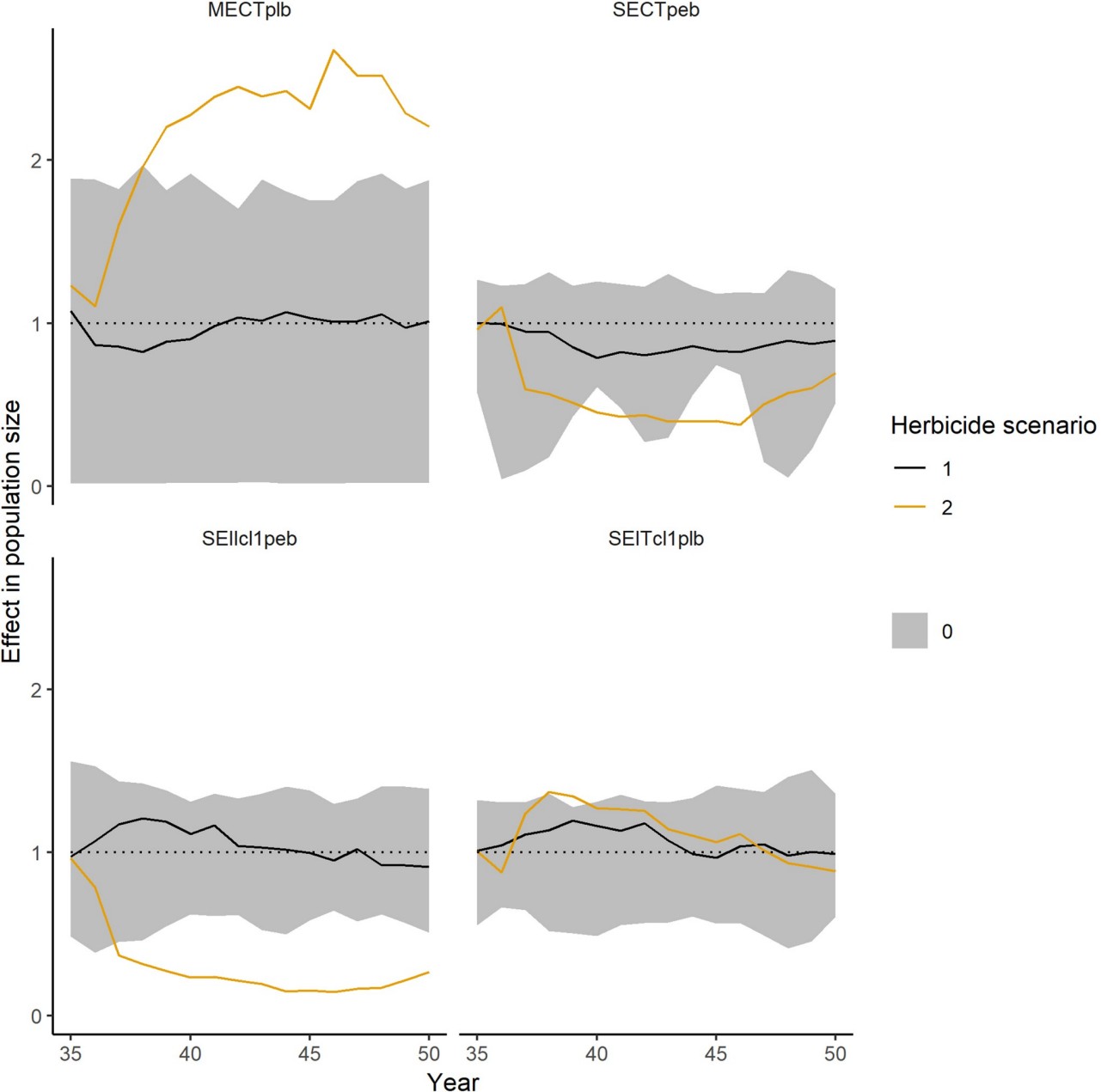

**Fig 7. Long-term impacts on the population sizes of selected PFTs over the simulated period.** Values below 1 represent a negative impact, values equal to 1 no impact and values above 1 a positive impact. Herbicide application starts in year 36 and ended in year 45. The theoretical herbicide had an impact on biomass and mortality. PFTs had random dose response curves. Grey ribbons show the fluctuations within control simulations (averaged for each year), the black lines show the mean (averaged for each year) of the 1.1 g a.i./ha application rate (herbicide scenario 1), the orange line the mean (averaged for each year) of the 3.3 g a.i./ha application rate (herbicide scenario 2).

During the first year of simulated herbicide application, the mean effect on the number of plant individuals exceeded 20% in 3 out of the 30 week growing period for the lower application rate (1.1 g a.i./ha, herbicide scenario 1) and the threshold of 50% in 5 out of 30 weeks growing period for the higher application rate (3.3 g a.i./ha, herbicide scenario 2) (Table 5). The inverse Simpson index had a negative mean effect of 20–30% in 5 of 30 weeks growing period for the herbicide scenario 1 (1.1 g a.i./ha) and exceeded the threshold of 50% in 5 of 30 weeks growing period for herbicide scenario 2 (3.3 g a.i./ha).

*Population-level.* The impact on population size and shoot mass is similar. Therefore, we only show the impact on population size for selected PFTs (Fig 7, see Table 2 for PFT code definitions). Only for the simulated herbicide application rate of 3.3 g a.i./ha (herbicide scenario 2), some PFTs showed long-term decreases in population sizes (SEIIcl1peb, SECTpeb, see Table 2 for PFT code definitions). For these PFTs, the mean effect is falling below the control range in some (SECTpeb) or all (SEIIcl1peb) years. In contrast, the PFT SEITcl1plb shows a slight increase in the mean population size (due to the decrease of interspecific competition, see [12] for further details), however it is only exceeding the control range in a few years. In contrast, the unfrequent PFT MECTplb, as indicated by a high variation within the control simulations representing high fluctuations in population size between the different MC runs, shows a strong increase in mean population size, even exceeding control variation. During the

**Table 6. Number of weeks in which the mean (minimal and maximal) negative effect on population size is within a certain effect class for two different PFTs.** As IBC-grass simulates only 30 weeks of growing period, the maximal number of weeks is 30. Simulated herbicide application started in year 36. Herbicide application rate of 1.1 g a.i./ha represents herbicide scenario 1 and the application rate of 3.3 g a.i./ha represents herbicide scenario 2. Numbers in brackets represent the number of weeks in which the minimal and maximal values are within a certain effect class. Please note, that only negative effects are considered in this table. Positive impacts can be observed in Figs 4–7.

| PFT | YEAR | APPLICATION RATE | <10% | 10–20% | 20–30% | 30–40% | 40–50% | >50% |
|---|---|---|---|---|---|---|---|---|
| MECTplb | 35 | 0 | 30 (0 0) | 0 (0 0) | 0 (0 0) | 0 (0 0) | 0 (0 0) | 0 (0 30) |
| | | 1.1 | 30 (0 0) | 0 (0 0) | 0 (0 0) | 0 (0 0) | 0 (0 0) | 0 (0 30) |
| | | 3.3 | 30 (0 0) | 0 (0 0) | 0 (0 0) | 0 (0 0) | 0 (0 0) | 0 (0 30) |
| | 36 | 0 | 30 (0 0) | 0 (0 0) | 0 (0 0) | 0 (0 0) | 0 (0 0) | 0 (0 30) |
| | | 1.1 | 18 (0 0) | 7 (0 0) | 0 (0 0) | 5 (0 0) | 0 (0 0) | 0 (0 30) |
| | | 3.3 | 24 (0 0) | 0 (0 0) | 1 (0 0) | 0 (0 0) | 0 (0 0) | 5 (5 30) |
| SECTpeb | 35 | 0 | 30 (0 0) | 0 (0 0) | 0 (0 14) | 0 (0 10) | 0 (0 5) | 0 (0 1) |
| | | 1.1 | 30 (0 0) | 0 (0 15) | 0 (0 9) | 0 (0 5) | 0 (0 1) | 0 (0 0) |
| | | 3.3 | 30 (0 0) | 0 (0 10) | 0 (0 10) | 0 (0 6) | 0 (0 4) | 0 (0 0) |
| | 36 | 0 | 30 (0 0) | 0 (0 0) | 0 (0 0) | 0 (0 10) | 0 (0 9) | 0 (0 11) |
| | | 1.1 | 25 (0 0) | 0 (0 0) | 5 (0 0) | 0 (0 1) | 0 (0 18) | 0 (0 11) |
| | | 3.3 | 23 (0 0) | 2 (0 1) | 0 (0 0) | 0 (0 0) | 0 (0 0) | 5 (5 9) |
| SEIIcl1peb | 35 | 0 | 30 (0 0) | 0 (0 0) | 0 (0 0) | 0 (0 0) | 0 (0 22) | 0 (0 8) |
| | | 1.1 | 30 (0 0) | 0 (0 0) | 0 (0 7) | 0 (0 14) | 0 (0 8) | 0 (0 1) |
| | | 3.3 | 29 (0 0) | 1 (0 0) | 0 (0 5) | 0 (0 14) | 0 (0 6) | 0 (0 5) |
| | 36 | 0 | 30 (0 0) | 0 (0 0) | 0 (0 0) | 0 (0 4) | 0 (0 6) | 0 (0 20) |
| | | 1.1 | 28 (0 0) | 2 (0 0) | 0 (0 0) | 0 (0 2) | 0 (0 16) | 0 (0 12) |
| | | 3.3 | 20 (0 0) | 1 (0 0) | 0 (0 1) | 0 (0 2) | 0 (0 5) | 9 (0 22) |
| SEITcl1plb | 35 | 0 | 30 (0 0) | 0 (0 0) | 0 (0 3) | 0 (0 13) | 0 (0 14) | 0 (0 0) |
| | | 1.1 | 30 (0 0) | 0 (0 0) | 0 (0 0) | 0 (0 3) | 0 (0 20) | 0 (0 7) |
| | | 3.3 | 30 (0 0) | 0 (0 0) | 0 (0 7) | 0 (0 19) | 0 (0 4) | 0 (0 0) |
| | 36 | 0 | 30 (0 0) | 0 (0 0) | 0 (0 14) | 0 (0 15) | 0 (0 1) | 0 (0 0) |
| | | 1.1 | 25 (0 0) | 4 (0 0) | 1 (0 0) | 0 (0 3) | 0 (0 13) | 0 (0 14) |
| | | 3.3 | 24 (0 2) | 0 (0 0) | 1 (0 0) | 0 (0 0) | 0 (4 13) | 5 (1 13) |

recovery period, starting in year 45, the population sizes of the PFTs SECTpeb and SEITcl1plb are falling back into the control variation. However, the PFTs MECTplb and SEIIcl1peb, are not able to recover within 5 years. All PFTs show no significant de- or increase in population sizes for the lower application rate (1.1 g a.i./ha, herbicide scenario 1).

Table 6 summarizes the negative effect magnitudes for the selected PFTs during the first year before herbicide application (year 35) and the first year of herbicide application (year 36). All PFTs show short-term impacts on the mean population size. For the low application rate of 1.1 g a.i./ha (herbicide scenario 1), only the PFT MECTplb shows an effect >30% for 5 weeks, the other three selected PFTs only show effects <30% for not more than 5 weeks per growing season. Under the higher application rate of 3.3 g a.i./ha (herbicide scenario 2), all selected PFTs show strong short-term impacts of >50% in 5–9 weeks.

## Conclusion

We presented a graphical user interface (GUI) of the plant community model IBC-grass to provide a user-friendly software tool, which simulates herbicide induced impacts on local non-target terrestrial plant communities. The GUI enhances previous console application of IBC-grass [12–14] by facilitating the application in herbicide risk assessments through guiding the user through the model parameter settings, analyses simulations and finally providing the user with a standardized graphical output. The software package is hosted as a GitHub repository, which is not only open access, but also open source (incl. the IBC-grass model source code, [16]). In this way, it is assured that it can be constantly reviewed and, consequently, improved and extended by the scientific community.

## Supporting information

**S1 File. Long-term validation of IBC-grass.**
(DOCX)

## Author Contributions

**Conceptualization:** Jette Reeg, Simon Heine, Christine Mihan, Sean McGee, Thomas G. Preuss, Florian Jeltsch.

**Formal analysis:** Jette Reeg.

**Methodology:** Jette Reeg.

**Project administration:** Simon Heine, Thomas G. Preuss, Florian Jeltsch.

**Resources:** Simon Heine, Christine Mihan, Sean McGee.

**Software:** Jette Reeg.

**Supervision:** Simon Heine, Thomas G. Preuss, Florian Jeltsch.

**Validation:** Jette Reeg, Simon Heine, Christine Mihan, Sean McGee, Thomas G. Preuss, Florian Jeltsch.

**Visualization:** Jette Reeg.

**Writing – original draft:** Jette Reeg.

**Writing – review & editing:** Jette Reeg, Simon Heine, Christine Mihan, Sean McGee, Thomas G. Preuss, Florian Jeltsch.

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
