## [Decision Letter · Decision Letter 0]

29 Aug 2019

PONE-D-19-21544

Herbicide risk assessments of non-target terrestrial plant communities: A graphical user interface for the plant community model IBC-grass

PLOS ONE

Dear Dr. Reeg,

Thank you for submitting your manuscript to PLOS ONE. After careful consideration, we feel that it has merit but does not fully meet PLOS ONE’s publication criteria as it currently stands. Therefore, we invite you to submit a revised version of the manuscript that addresses the points raised during the review process.

We would appreciate receiving your revised manuscript by Oct 13 2019 11:59PM. To enhance the reproducibility of your results, we recommend that if applicable you deposit your laboratory protocols in protocols.io, where a protocol can be assigned its own identifier (DOI) such that it can be cited independently in the future. For instructions see: http://journals.plos.org/plosone/s/submission-guidelines#loc-laboratory-protocols

We look forward to receiving your revised manuscript.

Kind regards,

Jen-Tsung Chen, Ph.D.

Academic Editor

PLOS ONE

Journal Requirements:

The project was funded by Bayer AG. Authors employed by Bayer AG worked on preparing this manuscript (see Author contributions).

We note that you received funding from a commercial source: Bayer AG.

Reviewers' comments:

Reviewer's Responses to Questions

**Comments to the Author**

1. Is the manuscript technically sound, and do the data support the conclusions?

Reviewer #1: Partly

Reviewer #2: Yes

2. Has the statistical analysis been performed appropriately and rigorously? 

Reviewer #1: Yes

Reviewer #2: N/A

3. Have the authors made all data underlying the findings in their manuscript fully available?

Reviewer #1: Yes

Reviewer #2: Yes

4. Is the manuscript presented in an intelligible fashion and written in standard English?

Reviewer #1: Yes

Reviewer #2: Yes

5. Review Comments to the Author

Reviewer #1: The manuscript “Herbicide risk assessments of non-target terrestrial plant communities: A graphical user interface for the plant community model IBC-grass” describes the applications of a graphical interface for the IBS-grass model, which evaluates the effects of herbicides over plant communities.

The manuscript is written in fluent English; nevertheless, it does not follow a clear structure to guide the reader in the analytical process. The Methods section is confused, and the Result section do not respond to the Introduction.

There is no explicit objective. This aspect conducts to reading difficulties. The subjacent objective is, in my opinion, to evaluate and validate a graphical interface of a preexistent model. In this way, there are many descriptions for the IBC-grass model that were already published elsewhere and should not be presented in detail here. Sometimes is not clear if the authors analyze the model or the graphical interface, which should be the main object of study.

The authors make a description of several ecological processes in the Methods section, but then in the results they do not explain the behavior of plant communities affected by the herbicides and shown in the figures. I recommend an appendix with all the ecological theory needed to support the results.

Data is fully available except for confidential data regarding dose response (in S1 file).

The authors do not explain how they manage to model the plant diversity, the herbicide diversity and all the possible interactions, without reaching an oversimplified result (see for example lines 240 to 246).

Some parameters, especially those from the figures are not explained, for example what is the meaning of the y axis in Figure 6. Some abbreviations have no reference.

Finally, the Conclusions section is not satisfactory. There are no true conclusions, but a synthesis of the background and references that should be part of the Introduction.

I recommend complete reworking.

Reviewer #2: The manuscript “Herbicide risk assessments of non-target terrestrial plant communities: A graphical user interface for the plant community model IBC-grass” (PONE-D-19-21544) introduces a graphical user interface (GUI) for the plant community model previously published and applied to herbicide risk assessment questions. Providing a GUI for a model intended for regulatory uses is greatly beneficial to its application by users not involved in the model development.

The manuscript is well written. It provides a clear and concise summary of the model itself and the GUI. An example application is described in detail, including the presentation of results on different temporal scales and levels of organization (community- and population-level impacts of an herbicide). In the supplemental material, a comparison of model outputs to published field-based empirical data are presented.

The model code (written in C++), the R-files for the GUI of the model along with the input files for the presented examples are available from GitHub. In addition, a comprehensive manual can be found on GitHub along with a documentation of the code. The only data set not available are the data for the dose-responses of glyphosate used as inputs to the simulations presented in the supplemental material.

The manuscript and the presented GUI for IBC-grass present a very useful contribution to the field of ecological risk assessment. The transparency of the model with its comprehensive documentation, availability and code documentation is exemplary. The GUI makes it accessible for use. Based on the review of the manuscript and the supplemental material, I recommend the manuscript for publication after addressing minor comments listed below.

However, there might be issues with the provided code on GitHub. I was not able to get the code running on my computer on Windows 10 (the MinGW installation was not recognized in command prompt). In addition, the R-package used (RGtk2Extras) is not supported/available for recent R versions (versions 3.5 and 3.6). My colleague tried to run the model under Windows 7, resulting in an error message from Rscript.exe about a missing file (‘libatk-1.0-0.dll’). I would be interested in trying the GUI and give feedback to the authors if they can provide help in getting it to run. It would be advisable to figure out the technical issues with model execution prior to publication.

Amelie Schmolke (schmolkea@waterborne-env.com)

Line-by-line comments:

Introduction

p. 3, l. 53-54: “As ecological models are limited by computational resources only, they overcome the spatial and temporal limitations as well as high resource requirements of empirical field studies.”

I would suggest to rephrase because models do have limitations other than computational ones.

p. 3, l. 60-61: “… models need to be validated against empirical data, proving that the simplified model is able to reflect real world conditions”

Needs to be rephrased as validation is not really a proof for anything. I recommend the following reference for insightful thoughts on model validation. Citing the reference would also help in the definition of ‘validation’ used in the current manuscript:

Rykiel EJ. 1996. Testing ecological models: The meaning of validation. Ecol Modell 90:229-244.

Methods

p. 6, Table 1: “BASED ON DATABASE” instead of “DATEBASE”

p. 7, l. 114: “IBS-grass” instead of “IBC-grass”

p. 9, l. 159: “… can be influenced by …”

The graphical user interface

p. 11, l. 201-202: A short description of what the environmental settings refer to would be helpful here.

p. 13, l. 233: “… the user has the possibility …”

p. 14, l. 261: “…, the user needs to …”

p. 14-15, l. 277-278: something seems to be missing in the sentence: “On the one had per time step, but also per year.”

Examples

p. 15, ‘Project settings’: it would be helpful to include a statement about the simulated herbicide application: was the herbicide applied once a year? What time of year?

Also, please include that a 5-yr recovery period was simulated after the simulation of herbicide effects.

Figure 3: include ‘herbicide scenario 1 / 2’ in the caption for each application rate (as done in the following figure captions)

Figures 5, 6: I assume the minimum of the yearly simulation outputs is shown rather than the weekly outputs as in figures 3 and 4? It would be good to mention that in the text and/or figure captions.

p. 18: in the paragraph ‘Population-level’, the authors need to refer back to Table 2, i.e. remind the reader of the codes for PFT (e.g., SECTpeb, etc.).

p. 19, l. 361: “negative effect frequencies” instead of “negative effect extends”?

p. 20, Table 6: Please clarify how the table should be read – how can the mean negative effect be <10% in all weeks, no minimal negative effect observed (always 0: I would expect the minimal effect to be <10% as well), but the maximal negative effect >50% in all 30 weeks? The calculation of the mean, minimal and maximal effects are described on p. 14, but it seems like I am missing something from that explanation.

Conclusions

p. 21, l. 391: “A thorough sensitivity analysis …”

The two paragraphs of the conclusions seem partly repetitive.

Supplemental material (S1 document)

The supplemental material presents interesting comparisons between empirical studies and model outputs. The document needs some proof reading: it includes a few typos, the figure numbers start at 6 and are not related to the figure references in the text, and there is a comment in the document that I assume is not meant for the reader.

I would suggest to refrain from stating “IBC-grass was validated with the long-term experimental data set …”, but rather refer to it as comparison of empirical data to model outputs or assessment of model performance. When the comparison would be considered as a successful validation was not discussed (in the main manuscript nor in the Supplemental material) and may be assessed differently by different readers. It would also be helpful to include a disclaimer that the assessment of the model performance is specific to the plant community and herbicide used in the empirical study.

6. PLOS authors have the option to publish the peer review history of their article (what does this mean?). If published, this will include your full peer review and any attached files.

Reviewer #1: Yes: Marcos Karlin

Reviewer #2: No

---

## [Author Response · Author response to Decision Letter 0]

8 Jan 2020

Detailed response to reviewers

Journal Requirements

Comment: When submitting your revision, we need you to address these additional requirements.

Response: We adapted the manuscript style to meet the style requirements of PLOS ONE. 

Comment: Please provide an amended Competing Interests Statement that explicitly states this commercial funder, along with any other relevant declarations relating to employment, consultancy, patents, products in development, marketed products, etc.

Response: The project was funded by Bayer AG. Authors employed by Bayer AG, namely SH, CM, TP, SM, worked on preparing this manuscript (see Author contributions). This does not alter our adherence to PLOS ONE policies on sharing data and materials. 

Reviewer 1

Comment: There is no explicit objective. This aspect conducts to reading difficulties. The subjacent objective is, in my opinion, to evaluate and validate a graphical interface of a preexistent model. In this way, there are many descriptions for the IBC-grass model that were already published elsewhere and should not be presented in detail here. Sometimes is not clear if the authors analyze the model or the graphical interface, which should be the main object of study.

Response: We have rewritten and clarified the objectives of the manuscript. 

“The plant community model IBC-grass (Individual-Based plant Community model for GRASSlands) represents such a suitable approach to extrapolate individual-level effects measured in standard guideline studies [5, 11] to plant populations in community context. Recent studies highlighted the capability of the model to detect herbicide induced impacts on plant communities [12], showed different sensitivities of important plant attributes [13] and validated IBC-grass against short-term [14] and long-term empirical data (see supporting information file S1 File). Model development is documented using the ODD protocol (Overview, Design concept and Details [15]). 

Although all requirements mentioned earlier are fulfilled, the model was, up to now, not convenient to use as it was developed as a console application, which would likely lead to hesitation to apply the model, especially for researchers not trained in modelling. Graphical user interfaces (GUI) are suitable tools to facilitate the application by guiding the user through settings and analyses of a simulation model. In this current study, we present a graphical user interface (GUI) for IBC-grass that facilitates the use of the model for risk assessment purposes without requiring any programming skills.” (p 4, l 67ff)

We partly restructured the methods of the manuscript: We included the flow chart of the processes included in the model (p 4, l 82ff; Figure 1). Only the main principles of IBC-grass are now explained within the main manuscript and we refer to the ODD-protocol in the supplemental material for detailed information on the implementation of the specific processes. We hope that this will clarify the method section and brings the focus of the manuscript to the graphical user interface.

Comment: Table 1: The table would be easier to interpret if lines are inserted between traits. Please reference the abbreviations and what is the meaning of the numbers.

Response: We have inserted lines between the traits to facilitate the interpretation of the table and included the trait life span. We included footnotes explaining values for the traits ecological strategy, grazing tolerance and symphenological groups. (p 6,Table 1)

Comment: L 118ff: Reference?

Response: We slightly rephrased the sentence and included a reference for the aboveground asymmetric competition for light resources: “Aboveground, only the size asymmetric competition for light is considered: taller plants with an erect growth form shade smaller plants growing as a rosette and thus acquire more resources [21].” (p 7, l 110f)

Comment: L 120ff: Reference? What happens with the root depth? And the alelopathy?

Response: We included an explanation, why we have simplified belowground competition: “For a model being a simplified version of the real world, belowground competition, on the other hand, is assumed to be size symmetric: resource distribution between plant individuals with overlapping zones of influences is independent of the root growth form and only depend on the root mass.” (p 7, l 112ff)

Comment: Shouldn't this section (The graphical user interface) be part of the results?

Response: We have shifted the description of the graphical user interface to the result section. (p 8, l 124ff)

Comment: L 239 ff: What about the herbicide type? Herbicides differ if they are systemic or for contact, differ on the active ingredient or the molecule... Do the model consider these aspects?

Response: IBC-grass does not directly account for different modes of actions. However, the user can select 6 different plant attributes being affected and besides vary the sensitivity of the plant functional type to cover for broad spectrum and selective herbicides. With these options, the mode of action can be indirectly addressed by affecting plant attributes. We rephrased the paragraph: “IBC-grass does not directly account for different modes of action. However, the user can select six different plant attributes to be affected by the herbicide, namely shoot mass, seedling shoot mass, survival, establishment, seed sterility and seed number, and vary the sensitivity of the different PFTs to cover for broad spectrum and selective herbicides. With these options, the mode of action can be indirectly addressed by affecting plant attributes.” (p 11, l 180ff)

Comment: L 259ff: This is a dangerous assumption. It depends on the crop, the wild species associated and the used herbicide (type, dose). A clear example is superweeds, and there is plenty of scientific evidence.

Response: The cited study of Christl et al. (2018) is an extensive literature review, which shows that there is in general no difference in sensitivity of crop vs. non-crop species. Super weeds are another topic as this is the term for weeds that have developed resistance against one or more herbicides and are thus hard to fight. In our model, we do not investigate super weeds. We rephrased the sentence: “However, the sensitivity of many non-crop plant species is still unknown, but the literature review by Christl et al. [29] comparing herbicide sensitivities of crop vs. weed species for various modes of action showed that the range of sensitivities of crop and non-crop species are comparable.” (p 12, l 198ff).

Comment: L 298ff: To what parameter do 0.7 respond?

Response: We included more details and gave an example. “i.e. a resulting standardized value 0.7 represents a 30% decrease in the specific endpoint compared to the mean of the control – e.g. in biomass” (p 13, l 236ff)

Comment: L 310: Please check the main subtitles according to the journal's guidelines.

Response: We checked that the main subtitles are according to the journal’s guidelines.

Comment: L 318: What does “initialization phase” mean?

Response: We included a short description. “The herbicide effect was simulated for 10 years after an initialization phase of 35 years where no application of any herbicide took place.” (p 14, l 256)

Comment: L 337: This example simulates only one year of herbicide application? Is the test different from that from Figure 5? (Note that Figure 5 is now Figure 6 as we included a flow chart)

Response: The Figure only shows the first year of simulated herbicide impact. But it is the same scenario as in Figure 6 (which shows the long-term impacts). We revised the figure caption: “Short-term impacts on different diversity indices during the first year of simulated herbicide application. Values below 1 represent a negative impact, values equal to 1 no impact and values above 1 a positive impact. The theoretical herbicide had an impact on biomass and mortality. PFTs had random dose response curves. Grey ribbons show the fluctuations within control simulations, the black lines show the mean for an application rate of 1.1 g a.i./ha (herbicide scenario 1) and the orange lines the mean for an application rate of 3.3 g a.i./ha (herbicide scenario 2).” (p 15, l 278ff)

Comment: L 338: The scale in the y axis is the fraction of the original average frequency (1)? Please explain in the text.

Response: We explained the scale in the caption of the figure. “Short-term impacts on number of plant individuals during the first year of simulated herbicide application. Values below 1 represent a negative impact, values equal to 1 no impact and values above 1 a positive impact. The theoretical herbicide had an impact on biomass and mortality. PFTs had random dose response-curves. Grey ribbon shows the fluctuation within control simulations, the black line shows the mean for an application rate of 1.1 g a.i./ha (herbicide scenario 1) and the orange line the mean for an application rate of 3.3 g a.i./ha (herbicide scenario 2).” (p 15, l 272ff)

Comment: L 355: Not clear if it is the same simulation than Fig 3 (now Fig 4) with other time scale or if it is a different simulation

Response: We included ‘short-term’ in the caption of Fig. 4+5.

Comment: Table 5: Numbers within brackets mean minimal and maximal? Their expression is not clear

Response: We included more details. “Numbers in brackets represent the number of weeks in which the minimal and maximal values are within a certain effect class. Please note, that only negative effects are considered in this table. Positive impacts can be observed in Fig. 4-7.” (p 16, l 307ff)

Comment: L 385ff: With the application of herbicide? Why?

Response: We included more explanation, but referred to Reeg et al. 2017 where indirect impacts of herbicides are explained in detail. “due to the decrease of interspecific competition, see [12] for further details” (p 17, l 315f)

Comment: L 387ff: Why?

Response: We gave a short clarification. “representing high fluctuations in population size between the different MC runs” (p 17, l 318f)

Comment: L 394ff: What does the y axis mean? Higher values indicate lesser or larger population size?

Response: We added the information in the figure caption “Long-term impacts on the population sizes of selected PFTs over the simulated period. Values below 1 represent a negative impact, values equal to 1 no impact and values above 1 a positive impact. Herbicide application starts in year 36 and ended in year 45. The theoretical herbicide had an impact on biomass and mortality. PFTs had random dose response curves. Grey ribbons show the fluctuations within control simulations (averaged for each year), the black lines show the mean (averaged for each year) of the 1.1 g a.i./ha application rate (herbicide scenario 1), the orange line the mean (averaged for each year) of the 3.3 g a.i./ha application rate (herbicide scenario 2).” (p 17, l 324ff)

As parts of the manuscript were removed during the revision process, some comments do not apply to the revised version anymore.

Reviewer 2

Comment: However, there might be issues with the provided code on GitHub. I was not able to get the code running on my computer on Windows 10 (the MinGW installation was not recognized in command prompt). In addition, the R-package used (RGtk2Extras) is not supported/available for recent R versions (versions 3.5 and 3.6). My colleague tried to run the model under Windows 7, resulting in an error message from Rscript.exe about a missing file (‘libatk-1.0-0.dll’). I would be interested in trying the GUI and give feedback to the authors if they can provide help in getting it to run. It would be advisable to figure out the technical issues with model execution prior to publication.

Response: As RGtk2Etras is not supported anymore in CRAN, we included the complete R software including all packages in our GitHub repository under Licence GPL (>= 3). Due to that change, the GUI package is now only applicable for Windows OS. However, the underlying IBCgrass model can be nevertheless compiled and run under Linux OS and Mac OS.

As stated in the requirements, the g++ compiler needs to be set as environmental variable (see Manual for further details). We have tested the package again under Windows 10 and got no errors. 

Please let us know in case you are still facing difficulties running the program.

Comment: p. 3, l. 53-54: “As ecological models are limited by computational resources only, they overcome the spatial and temporal limitations as well as high resource requirements of empirical field studies.”

I would suggest to rephrase because models do have limitations other than computational ones.

Response: We rephrased the sentence to: “Ecological models overcome the spatial and temporal limitations as well as high resource requirements of empirical field studies.” (p 3, l 53ff)

Comment: p. 3, l. 60-61: “… models need to be validated against empirical data, proving that the simplified model is able to reflect real world conditions”

Needs to be rephrased as validation is not really a proof for anything. I recommend the following reference for insightful thoughts on model validation. Citing the reference would also help in the definition of ‘validation’ used in the current manuscript:

Rykiel EJ. 1996. Testing ecological models: The meaning of validation. Ecol Modell 90:229-244.

Response: We rephrase the sentence to: ““Therefore ecological models need to fulfil certain requirements to be considered as suitable higher tier approaches: (1) comparison of model predicted effects against empirically measured data to increase the credibility of the simplified model to realistically reflect herbicide impacts [7, 8];” (p 3, l 58ff)

Comment: p. 6, Table 1: “BASED ON DATABASE” instead of “DATEBASE”

Response: We changed it.

Comment: p. 7, l. 114: “IBS-grass” instead of “IBC-grass”

Response: We changed it.

Comment: p. 9, l. 159: “… can be influenced by …”

Response: We changed it.

Comment: p. 11, l. 201-202: A short description of what the environmental settings refer to would be helpful here.

Response: We added “(abiotic as well as biotic conditions such as resource availability and biotic disturbances)” (p 10, l 156ff).

Comment: p. 13, l. 233: “… the user has the possibility …”

Response: We changed it.

Comment: p. 14, l. 261: “…, the user needs to …”

Response: We changed it.

Comment: p. 14-15, l. 277-278: something seems to be missing in the sentence: “On the one had per time step, but also per year.”

Response: We revised the sentence and changed it to: “per time step as well as per year” (p 13, l 240)

Comment: p. 15, ‘Project settings’: it would be helpful to include a statement about the simulated herbicide application: was the herbicide applied once a year? What time of year?

Also, please include that a 5-yr recovery period was simulated after the simulation of herbicide effects.

Response: We added the missing information: “(1 application/year, in the first week of the growing period)” (p 14, l 256)

Comment: Figure 3: include ‘herbicide scenario 1 / 2’ in the caption for each application rate (as done in the following figure captions)

Response: We included it in all figure captions.

Comment: Figures 5, 6: I assume the minimum of the yearly simulation outputs is shown rather than the weekly outputs as in figures 3 and 4? It would be good to mention that in the text and/or figure captions.

Response: We adapted the captions according to the comments. “Grey ribbons show the fluctuations within control simulations (averaged over each year), the black lines show the mean (averaged over each year) of the 1.1 g a.i./ha application rate (herbicide scenario 1), the orange line the mean (averaged over each year) of the 3.3 g a.i./ha application rate (herbicide scenario 2).” (p 165, l 289ff) The mean gives, due to variability, a better overview of modelling results.

Comment: p. 18: in the paragraph ‘Population-level’, the authors need to refer back to Table 2, i.e. remind the reader of the codes for PFT (e.g., SECTpeb, etc.).

Response: We referred back to Table 2: “see Table 2 for PFT code definitions” (p 17, l 311)

Comment: p. 19, l. 361: “negative effect frequencies” instead of “negative effect extends”?

Response: We replaced “effect frequencies” with “effect magnitudes” (p 17, l 331).

Comment: p. 20, Table 6: Please clarify how the table should be read – how can the mean negative effect be <10% in all weeks, no minimal negative effect observed (always 0: I would expect the minimal effect to be <10% as well), but the maximal negative effect >50% in all 30 weeks? The calculation of the mean, minimal and maximal effects are described on p. 14, but it seems like I am missing something from that explanation.

Response: We included additional information in the caption: “Herbicide application rate of 1.1 g a.i./ha represents herbicide scenario 1 and the application rate of 3.3 g a.i./ha represents herbicide scenario 2. Numbers in brackets represent the number of weeks in which the minimal and maximal values are within a certain effect class. Please note, that only negative effects are considered in this table. Positive impacts can be observed in Fig. 4-7.” (p 18, l 338ff).

Comment: p. 21, l. 391: “A thorough sensitivity analysis …”

The two paragraphs of the conclusions seem partly repetitive.

Response: We rewrote the paragraph. “We presented a graphical user interface (GUI) of the plant community model IBC-grass to provide a user-friendly software tool, which simulates herbicide induced impacts on local non-target terrestrial plant communities. The GUI enhances previous console application of IBC-grass [12-14] by facilitating the application in herbicide risk assessments through guiding the user through the model parameter settings, analyses simulations and finally providing the user with a standardized graphical output. The software package is hosted as a GitHub repository, which is not only open access, but also open source (incl. the IBC-grass model source code, [16]). In this way, it is assured that it can be constantly reviewed and, consequently, improved and extended by the scientific community.“ (p 18, l 347ff)

Comment: Supplemental material (S1 document): The document needs some proof reading: it includes a few typos, the figure numbers start at 6 and are not related to the figure references in the text, and there is a comment in the document that I assume is not meant for the reader. I would suggest to refrain from stating “IBC-grass was validated with the long-term experimental data set …”, but rather refer to it as comparison of empirical data to model outputs or assessment of model performance. When the comparison would be considered as a successful validation was not discussed (in the main manuscript nor in the Supplemental material) and may be assessed differently by different readers. It would also be helpful to include a disclaimer that the assessment of the model performance is specific to the plant community and herbicide used in the empirical study.

Response: We revised the document according to typos and figure numbers. As the reviewer suggested we changed the term validation to ‘comparison of empirical data to model predictions’ or ‘model performance’.

---

## [Decision Letter · Decision Letter 1]

5 Feb 2020

PONE-D-19-21544R1

Herbicide risk assessments of non-target terrestrial plant communities: A graphical user interface for the plant community model IBC-grass

PLOS ONE

Dear Dr. Reeg,

Thank you for submitting your manuscript to PLOS ONE. After careful consideration, we feel that it has merit but does not fully meet PLOS ONE’s publication criteria as it currently stands. Therefore, we invite you to submit a revised version of the manuscript that addresses the points raised during the review process.

We would appreciate receiving your revised manuscript by Mar 21 2020 11:59PM. To enhance the reproducibility of your results, we recommend that if applicable you deposit your laboratory protocols in protocols.io, where a protocol can be assigned its own identifier (DOI) such that it can be cited independently in the future. For instructions see: http://journals.plos.org/plosone/s/submission-guidelines#loc-laboratory-protocols

We look forward to receiving your revised manuscript.

Kind regards,

Jen-Tsung Chen, Ph.D.

Academic Editor

PLOS ONE

Reviewers' comments:

Reviewer's Responses to Questions

**Comments to the Author**

1. If the authors have adequately addressed your comments raised in a previous round of review and you feel that this manuscript is now acceptable for publication, you may indicate that here to bypass the “Comments to the Author” section, enter your conflict of interest statement in the “Confidential to Editor” section, and submit your "Accept" recommendation.

Reviewer #1: (No Response)

Reviewer #2: All comments have been addressed

2. Is the manuscript technically sound, and do the data support the conclusions?

Reviewer #1: Yes

Reviewer #2: (No Response)

3. Has the statistical analysis been performed appropriately and rigorously? 

Reviewer #1: Yes

Reviewer #2: (No Response)

4. Have the authors made all data underlying the findings in their manuscript fully available?

Reviewer #1: Yes

Reviewer #2: (No Response)

5. Is the manuscript presented in an intelligible fashion and written in standard English?

Reviewer #1: Yes

Reviewer #2: (No Response)

6. Review Comments to the Author

Reviewer #1: The manuscript “Herbicide risk assessments of non-target terrestrial plant communities: A graphical user interface for the plant community model IBC-grass” describes the applications of a graphical interface for the IBS-grass model, which evaluates the effects of herbicides over plant communities.

Authors have replied adequately almost all comments from the original review.

The manuscript is written in fluent English; the structure has been improved respect the original manuscript.

Still, the objective is not explicit. However, some new paragraphs insinuate the real objective of the manuscript.

Some aspects of the modeling are oversimplified, however it may be used in specific applications as authors indicate in the manuscript.

Finally, the Conclusions section has been now corrected.

I recommend minor revision.

Reviewer #2: The authors addressed all comments adequately. I think the paper looks very good now, and will be a useful guide to the presented GUI for the IBC-grass model.

I downloaded the updated repository files for the interface available from GitHub, and can now run the model without issues.

7. PLOS authors have the option to publish the peer review history of their article (what does this mean?). If published, this will include your full peer review and any attached files.

Reviewer #1: Yes: Marcos Sebastián Karlin

Reviewer #2: No

---

## [Author Response · Author response to Decision Letter 1]

19 Feb 2020

We would like to thank both reviewers for their comments. As reviewer #2 had no further comments, we only response to the comments raised by the reviewer #1.

Reviewer #1

Comment: L 78: I suppose this is the objective. Please, explicit it: "The objective of this paper is..."

Response: We changed the sentence accordingly. (L 78)

Comment: Figure 1, L 86ff: Orange and blue colours are not seen in Fig 1

Response: We corrected the figure caption (L86 ff): “Grey boxes indicate processes occurring in each simulated week, green boxes indicate processes occurring only in specific weeks and blue boxes indicate potential herbicide-induced effects that the user can turn on or off. Striped boxes indicate processes the user can adjust and change.”

Comment: Table 2 (L 103): The original comment on Table 2 was not answered: "this is not in Table 1. I guess, perennial: cycle more than 2 years long; annual: cycle 1 year long. No biennials?"

Response: We have included the trait in Table 1 (L 101). Annuals are indeed defined as plants living for 1 year; perennials are defined as plants living for 100 years. Thus, biennials are not included.

---

## [Editor Report · Decision Letter 2]

20 Feb 2020

Herbicide risk assessments of non-target terrestrial plant communities: A graphical user interface for the plant community model IBC-grass

PONE-D-19-21544R2

Dear Dr. Reeg,

We are pleased to inform you that your manuscript has been judged scientifically suitable for publication and will be formally accepted for publication once it complies with all outstanding technical requirements.

With kind regards,

Jen-Tsung Chen, Ph.D.

Academic Editor

PLOS ONE
---

## [Editor Report · Acceptance letter]

28 Feb 2020

PONE-D-19-21544R2 

Herbicide risk assessments of non-target terrestrial plant communities: A graphical user interface for the plant community model IBC-grass 

Dear Dr. Reeg:

I am pleased to inform you that your manuscript has been deemed suitable for publication in PLOS ONE. Congratulations! Your manuscript is now with our production department. 

With kind regards,

on behalf of

Dr. Jen-Tsung Chen 

Academic Editor

PLOS ONE